# A shared core microbiome in soda lakes separated by large distances

Jackie K. Zorz [1], Christine Sharp [1], Manuel Kleiner[2], Paul M.K. Gordon [3], Richard T. Pon[3], Xiaoli Dong[1] & Marc Strous [1]

In alkaline soda lakes, concentrated dissolved carbonates establish productive phototrophic microbial mats. Here we show how microbial phototrophs and autotrophs contribute to this exceptional productivity. Amplicon and shotgun DNA sequencing data of microbial mats from four Canadian soda lakes indicate the presence of > 2,000 species of Bacteria and Eukaryotes. We recover metagenome-assembled-genomes for a core microbiome of < 100 abundant bacteria, present in all four lakes. Most of these are related to microbes previously detected in sediments of Asian alkaline lakes, showing that common selection principles drive community assembly from a globally distributed reservoir of alkaliphile biodiversity. Detection of > 7,000 proteins show how phototrophic populations allocate resources to specific processes and occupy complementary niches. Carbon fixation proceeds by the Calvin-Benson-Bassham cycle, in Cyanobacteria, Gammaproteobacteria, and, surprisingly, Gemmatimonadetes. Our study provides insight into soda lake ecology, as well as a template to guide efforts to engineer biotechnology for carbon dioxide conversion.

[1] Department of Geoscience, University of Calgary, Calgary, AB T2N 1N4, Canada. [2] Department of Plant and Microbial Biology, North Carolina State University, Raleigh, NC 27695, USA. [3] Centre for Health Genomics and Informatics, University of Calgary, Calgary, AB T2N 2T9, Canada. Correspondence and requests for materials should be addressed to J.K.Z. (email: jacqueline.zorz@ucalgary.ca)

Soda lakes are among the most alkaline natural environments on earth, as well as among the most productive aquatic ecosystems known[1,2]. The high productivity of soda lakes is due to a high bicarbonate concentration. Tens to hundreds of millimolars of bicarbonate are typically available for photosynthesis using carbon concentrating mechanisms[3,4], compared to generally <2 mM in the oceans[5]. This can lead to the formation of thick, macroscopic microbial mats with rich microbial biodiversity[6]. Because of the high pH, alkalinity, and high sodium salinity of these environments, the microorganisms that reside in soda lakes are considered extremophiles[7]. Using conditions of high pH and alkalinity is also a promising option to improve the cost-effectiveness of biotechnology for biological carbon dioxide capture and conversion[8–10].

Soda lakes have contributed to global primary productivity on a massive scale in Earth's geological past[11]. Currently, groups of much smaller soda lakes exist, for example, in the East African Rift Zone, rain-shadowed regions of California and Nevada, and the Kulunda steppe in South Russia[12]. Many microorganisms have been isolated from these lakes. These include cyanobacteria[13–15], chemolithoautotrophic sulfide oxidizing bacteria[16–18], sulfate reducers[19,20], nitrifying[21,22], and denitrifying bacteria[23], as well as aerobic heterotrophic bacteria[24,25], methanotrophs[26], fermentative bacteria[27,28], and methanogens[29]. Recently, almost one thousand metagenome-assembled-genome sequences (MAGs) were obtained from sediments of Kulunda soda lakes[30].

In the present study we investigate the microbial mat community structure of four alkaline soda lakes located on the Cariboo Plateau in British Columbia, Canada. This region has noteworthy geology and biology due to the diversity in lake brine compositions within a relatively small region[31]. There are several hundred shallow lakes on the Cariboo Plateau and these range in size, alkalinity, and salinity. Underlying basalt in some areas of the plateau, originating from volcanic activity during the Miocene and Pliocene eras, offers ideal conditions for forming soda lakes, as it provides little soluble calcium and magnesium[6,32,33]. Some of these lakes harbor seasonal microbial mats that are either dominated by cyanobacteria or eukaryotic green algae. However, beyond this little is currently known about these systems in terms of microbiology.

We use a combination of shotgun metagenomes, and 16 S and 18 S rRNA amplicon sequencing to establish a microbial community structure for the microbial mats of four soda lakes. We perform proteomics to show how specific populations allocate resources to specific metabolic pathways, focusing on photosynthesis, and carbon, nitrogen, and sulfur cycles. Through the use of metagenomics and metaproteomics, this study provides a comprehensive molecular characterization of a phototrophic microbial mat microbiome. Specifically, we offer evidence in support of widespread phototrophy and niche differentiation among populations inhabiting these alkaline microbial mats, as well as the unexpected potential for mixotrophy in a member of the Gemmatimonadetes phylum. Also, by comparing metagenomic reads between the present study and a recent study from soda lake sediments 8000 km away in central Asia[30], we find the presence of a core soda lake microbiome with some strikingly similar populations, potentially the result of recent dispersal events.

## Results and discussion

### Soda lake geochemistry and community composition.
The Cariboo Plateau contains hundreds of lakes of different size, alkalinity and salinity. Here we focused on four alkaline soda lakes (Fig. 1) that feature calcifying microbial mats with similarities to ancient stromatolites or thrombolites[6,34,35]. Between 2014 and 2017, the total alkalinity in these lakes was between 0.20–0.65 mol $L^{-1}$ at pH 10.1–10.7 (Supplementary Table 1). Four years of amplicon sequencing data (16 S and 18 S rRNA) showed the microbial mats contain at least 1662 bacterial and 587 eukaryotic species-level operational taxonomic units (OTUs, clustered at 97% similarity) (Supplementary Data 1). The mat communities from different lakes were similar, but distinct, and relatively stable over time (Fig. 1). Probe, Deer and Goodenough Lakes harbored predominantly cyanobacterial mats, whereas the mats of more saline Last Chance Lake contained mainly phototrophic Eukaryotes. This was shown with proteomics (see below), because it was impossible to compare abundances of Eukaryotes and Bacteria using amplicon sequencing. Bacterial species associated with 340 OTUs were found in all four lakes. These species accounted for 20.5% of the region's species richness and 84% of the total sequenced reads, suggesting that there is a common and abundant core microbiome shared among the alkaline lakes of the Cariboo Plateau. Despite the high proportion of eukaryotic biomass and phototrophs, the core alkaline lake, prokaryotic microbiome was still present in Last Chance Lake (although at lower relative abundances).

### Metagenomes reveal similarity between distant lakes.
After amplicon sequencing had outlined the core microbiome of the Cariboo soda lake microbial mats, shotgun metagenome sequencing, assembly, and binning were used to obtain the provisional whole-genome sequences, or metagenome-assembled-genomes (MAGs), of its key microbiota. We selected 91 representative, de-replicated MAGs for further analysis (Supplementary Data 2). Most of these MAGs were near-complete (>90% for 85 MAGs), and contained relatively few duplicated conserved single-copy genes (<5%, for 83 MAGs). For fifty-six MAGs, we independently assembled and binned 2–5 nearly identical (>95% average nucleotide identity) versions, indicating the presence of multiple closely related strains. 40–60% of quality-controlled reads were mapped to the 91 MAGs, showing that the associated bacteria accounted for approximately half of the DNA extracted. Most of the remaining reads were mapped to MAGs of lower quality and coverage, associated with a much larger group of less abundant bacteria. This was not surprising because amplicon sequencing had already indicated the presence of >2000 different bacterial and eukaryotic OTUs. Full length 16 S rRNA gene sequences (Supplementary Data 3) were reconstructed from shotgun metagenome reads. Fifty-seven of those could be associated with a MAG based on taxonomic classifications and abundance profiles. Perfect alignment of full length 16 S rDNA gene sequences to consensus OTU amplicon sequences showed that almost all these MAGs were core Cariboo microbiome members, present in each lake (Supplementary Table 2).

Figure 2 shows the taxonomic affiliation and average relative sequence abundances for the bacteria associated with the MAGs. For taxonomic classification we used the recently established GTDB taxonomy[36]. We also used the GTDB toolkit to investigate the similarity of the Cariboo mat genomes to >800 MAGs recently obtained from sediments of the Central Asian soda lakes of the Kulunda Steppe[30]. The distance between the two systems of alkaline lakes is approximately 8000 km. Yet, 56 of the Cariboo MAGs were clustered together with Kulunda MAGs and defined family or genus level diversity in the context of the GTDB database (release 86, >22,000 whole-genome sequences). This degree of similarity between geographically distant lake systems was surprising, especially because DNA was obtained from Kulunda sediments, not mats. It suggests that the core microbiome defined here for Cariboo lake mats, also applies to at least one other, well described system of soda lakes.

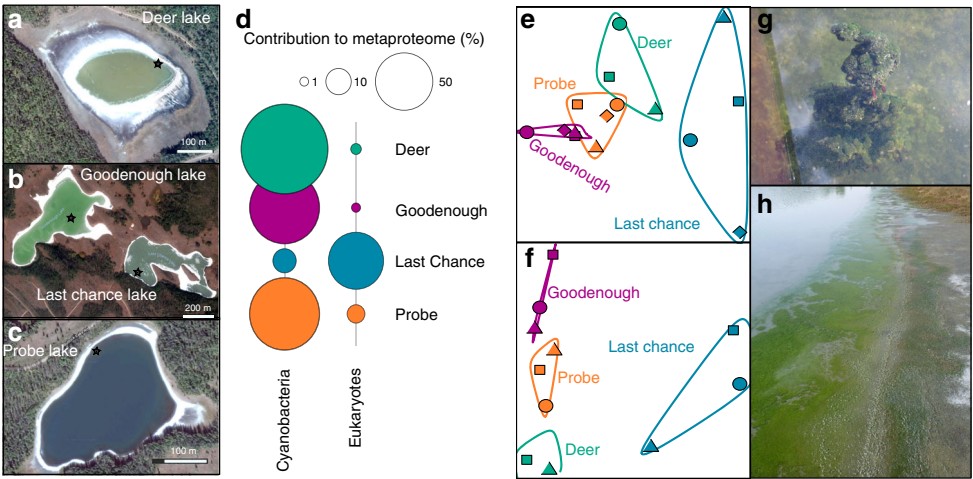

**Fig. 1** Images and biological description of Cariboo Plateau soda lakes. Google Satellite images of **a** Deer Lake, **b** Goodenough and Last Chance Lakes, and **c** Probe Lake. Black stars indicate approximate sampling locations. **d** Bubble plots showing the relative contribution of Cyanobacteria and Eukaryotes to the lake metaproteomes. **e** Non-metric multidimensional scaling (NMDS) plots using Bray-Curtis dissimilarity to visualize the microbial communities of the soda lake mats over years of sampling using 16 S rRNA amplicon sequencing data, and **f** 18 S rRNA amplicon sequencing data. Shapes indicate year of sampling: Circles: 2014, square: 2015, diamond: 2016, triangle: 2017. Samples for 18 S rRNA analysis were not taken in 2016, and Deer Lake samples were not taken in 2014 for 18 S, and 2016 for 16 S. NMDS Stress values were below 0.11. **g** Image of microbial mat from Goodenough Lake, and **h** Image of microbial mats along the shore of Last Chance Lake. Map data: Google, Maxar Technologies

Interestingly, the genetic distance between the most similar MAGs from each of the two regions decreased with increasing abundance in Cariboo mats (Pearson correlation −0.49, p: 0.0003, $n = 48$, Fig. 2b, Supplementary Data 2), but not with abundance in Kulunda sediments. For example, the most abundant Cariboo cyanobacterium (C1—affiliated with *Nodosilinea*, relative abundance >7%) displayed 99% average nucleotide identity over 85% of its genome with Kulunda MAG GCA_003550805. The latter displayed <0.1% relative abundance in Kulunda sediments. Mapping of Kulunda sequencing reads directly to Cariboo genomes (Supplementary Data 2) did not provide any evidence for the presence of previously undetected bacteria/MAGs in Kulunda sediments that were more similar to Cariboo bacteria/MAGs than those presented by Vavourakis et al.[30].

These results suggest that when the Cariboo lakes formed ~10,000 years ago after the last ice age[6], their microbiomes assembled from a much older, global reservoir of alkaliphile biodiversity. The striking relationship between Cariboo abundance and Kulunda-Cariboo relatedness might be explained by increased rates of successful dispersal/colonization for more abundant populations. Identification of vectors for dispersal still awaits future research, but bird migration is an obvious candidate. For example, the Northern Wheatear, which migrates between Northern Canada and Africa via Central Asia, could potentially link many known soda lakes worldwide. Abundance in sediments, located below mats, might not explain dispersal well, because sediments are less exposed to dispersal vectors than mats.

In any case, the genetic distances separating related bacteria were generally large, indicating that successful colonization by invading bacteria from a different lake system must be extremely rare. Possibly, only a single bacterium (MAG C1) traveled between and successfully colonized another lake system since the last ice age. A strong degree of isolation was also observed for other ecological islands, such as hot springs[37]. Thus, the observed similarities of the microbiota between distant lake systems indicate shared outcomes of community assembly for microbial mat microbiomes in two distant soda lake environments. Future studies will indicate whether the core microbiota of Kulunda and Cariboo soda lakes has also assembled in other soda lakes.

Dispersal between Cariboo soda lakes, separated by at most 40 km, was very effective. For all 56 sets of 2–5 nearly identical MAG variants (average nucleotide identity >95%) we detected co-occurrence of all variants (Supplementary Data 4). This also showed that competitive exclusion was irrelevant, even for these nearly identical bacteria. Comparison of ratios of synonymous and non-synonymous mutations among the most rapidly evolving core genes—genes present in all genome variants, Supplementary Data 5—showed that diversifying selection acted on 775 genes, including many transporters and genes involved in cell envelope biogenesis. Accessory genes—not encoded on all variant genomes—and CRISPRs could display many more ecologically relevant differences, which could prevent competitive exclusion.

**Proteomics reveals niche partitioning of cyanobacteria**. The processes that dictate assembly of effective phototrophic microbial mat communities are well understood, with ecological adaptations and responses to dynamic light, oxygen, sulfide, pH, and carbon dioxide gradients[38]. But, to what extent do these known rules of engagement also apply to alkaline soda lake microbial mats, where primary productivity has access to unlimited inorganic carbon[2], as was previously shown for Cariboo Soda lakes[6]? We performed environmental proteomics and connected protein expression to abundant MAGs to answer this question for the Cariboo Plateau soda lake mats (Supplementary Data 6).

Over 7000 expressed proteins were identified, with high confidence, in daytime mat samples from each of the lakes. For comparison, the most comprehensive environmental proteomes obtained so far have identified up to ~10,000 proteins[39]. Given the high diversity and extremely complex nature of the mat samples, identification of 7217 proteins is an excellent starting point for ecophysiological interpretation. Approximately half of the expressed proteins could be attributed to the 91 MAGs, consistent with abundance estimates inferred from amplicon and shotgun data. This enabled us to investigate how the bacteria associated with the MAGs distributed their resources over different ecophysiological priorities[40]. Given that a substantial

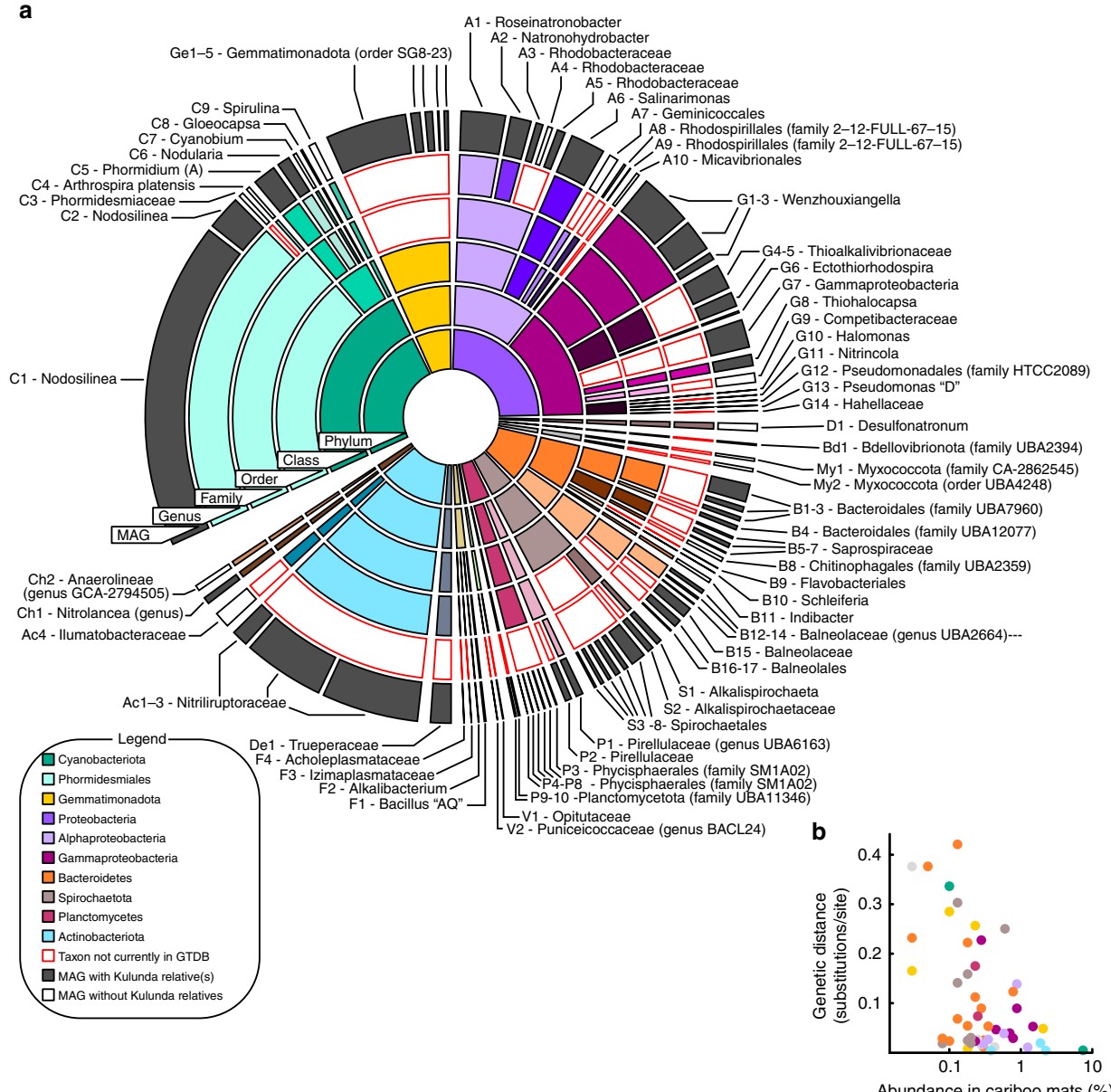

**Fig. 2** Relative abundance and diversity of soda lake metagenome-assembled-genomes. **a** Sunburst diagram showing relative abundances and GTDB taxonomic classifications of metagenome-assembled-genomes (MAGs) obtained from Cariboo lakes. Core-microbiome MAGs with closest relatives among Central Asian (Kulunda) soda lake MAGs are shown in grey. Red outlines indicate new clades that were not yet represented in GTDB. For example, MAG C1, the most abundant MAG, is affiliated with the genus *Nodosilinea*, which was represented in GTDB, with a Kulunda MAG more similar than any genome present in GTDB. **b** Scatter plot showing for each core microbiota the genetic distance between Cariboo and Kulunda representatives as a function of the abundance in Cariboo mat samples. This relationship is statistically significant (Pearson's correlation r: −0.49, n = 48, p < 0.05), but no such relationship was detected for the abundance of Kulunda MAGs. See also Supplementary Data 2

amount of cellular energy goes towards manufacturing proteins, the relative proportion of a proteome dedicated to a particular function provides an estimate of how important that function is to the organism. Proteomic data were also used to estimate the $^{13}C$ content of some abundant species, providing additional information on which carbon source they used and to what extent their growth was limited by carbon availability[41]. Brady et al. (2013) previously showed that microbial mat organic matter had $\delta^{13}C$ values of −19 to −25‰, up to 27‰ depleted in $^{13}C$ compared to bulk dissolved carbonates, consistent with non-$CO_2$-limited photosynthesis[6]. Overall protein $\delta^{13}C$ values for the four lakes inferred from the proteomics data in the present study were

between −19 and −25‰, in line with previous results for mat organic matter.

Consistent with their reputation as productive ecosystems with virtually unlimited access to inorganic carbon, the most abundant bacteria were large, mat-forming (filamentous) cyanobacteria, related to *Nodosilinea* and *Phormidium*. Pigment antenna proteins and photosynthetic reaction center proteins accounted for the largest fraction of detected proteins overall. The organism with the highest presence in the metaproteome was the cyanobacterial MAG C1, affiliated with *Nodosilinea* and accounting for up to 42% of mat metaproteomes. Remarkably, we were able to identify 1103 proteins from this MAG, 27% of its predicted proteome (Fig. 3).

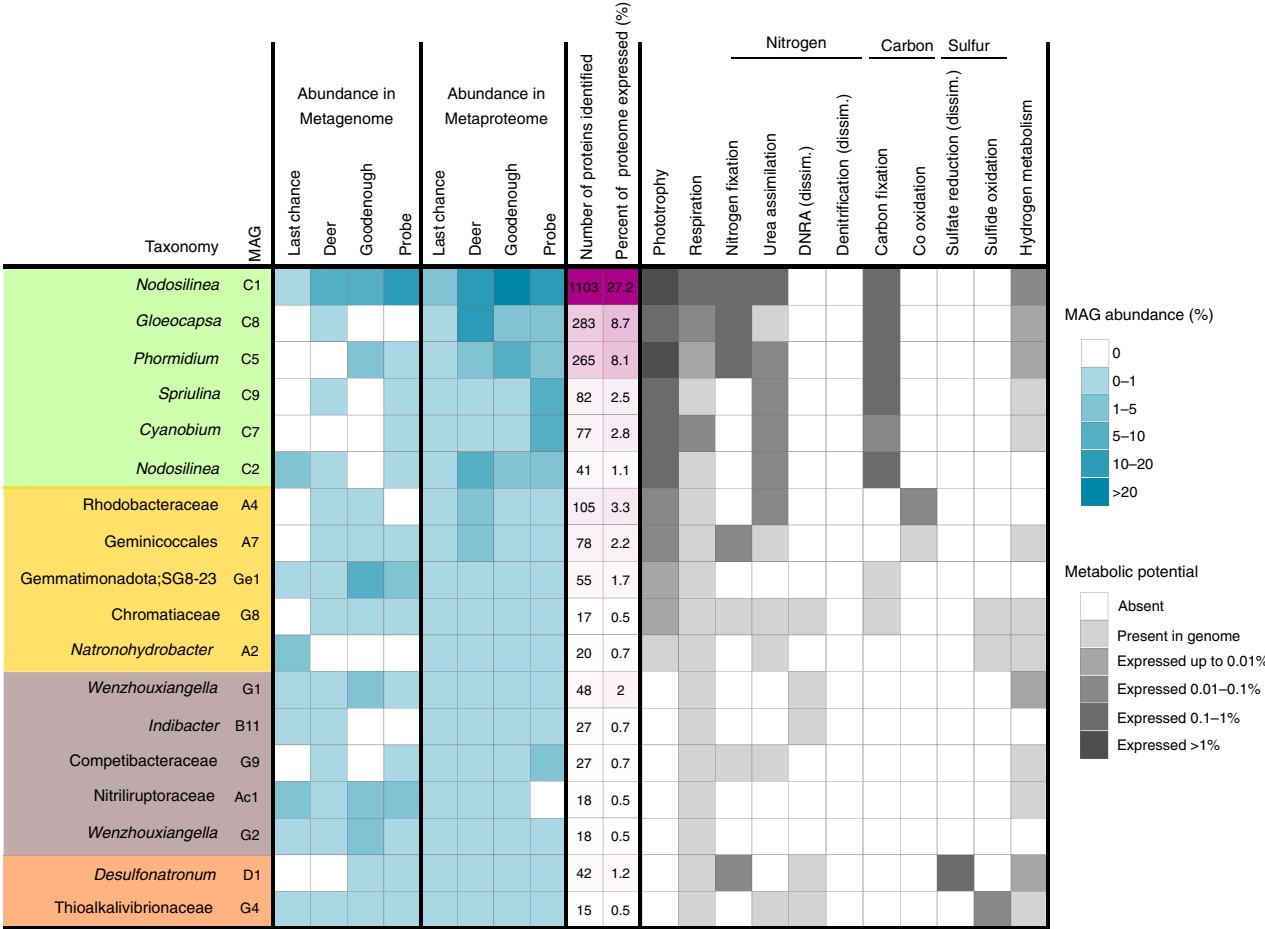

**Fig. 3** Summary of metabolism and protein expression of soda lake populations. Heatmap showing abundances and expressed functions for metagenome-assembled-genomes (MAGs) with at least 15 proteins identified in the metaproteomes. MAGs are broadly arranged based on function, with photoautotrophs in green, anoxygenic phototrophs in yellow, sulfur cycling in orange, and other heterotrophic bacteria in brown. Metabolic potential was inferred from the genes listed in Supplementary Data 6. If the gene was identified in a metaproteome it was considered expressed, and is shaded according to its highest relative abundance (% of all peptide spectral matches) in the four lake metaproteomes. MAG abundances in metagenome and metaproteome were estimated as explained in Materials and methods

This level of detection is comparable to the proteomics results from other studies of pure cultures of cyanobacteria, such as Arthrospira, 21%, and Cyanothece, 47%[42,43]. Nine cyanobacterial MAGs were assembled in total, and proteins from all nine were detected in the metaproteomes of all four lakes (Fig. 3, Supplementary Data 6). It is clear that the presence of so many cyanobacteria provides functional redundancy and contributes to functional robustness and resiliency[44,45]. However, we also detected strong evidence for niche differentiation for those cyanobacteria with larger numbers of proteins detected, in particular MAG C1 (*Nodosilinea*), and MAG C5 (*Phormidium A*) (Fig. 4).

Phycobilisomes, the large, proteinaceous, light harvesting complexes of cyanobacteria, contain an assortment of pigments, which absorb at different wavelengths of light, and re-emit that light at longer wavelengths, around 680 nm, compatible with the reaction center of Photosystem II. Phycobilisome pigment composition varied among the cyanobacterial populations, leading to niche differentiation based on light quality, as was also observed in the marine environment[46]. C1 and most other cyanobacterial populations expressed high amounts of phycocyanin, maximum absorbance 620 nm, and allophycocyanin, maximum absorbance 650 nm. In contrast, C5 uniquely expressed the pigment phycoerythrocyanin, with a maximum absorbance at 575 nm (Fig. 4). Phycoerythrocyanin would enable this

population to absorb shorter wavelengths of light, in comparison to its cyanobacterial neighbours, and expands the spectral reach of photosynthesis for these mat communities, increasing productivity. The absence of expression of phycoerythrin, which has a maximum absorbance at 495 and 560 nm, is consistent with the light attenuation profile of aquatic environments with high dissolved organic matter, such as productive alkaline lakes, where wavelengths <500 nm are rapidly attenuated[47,48].

Shorter wavelength light (blue/green light) has higher energy, and high energy photons can damage photosynthetic machinery in cyanobacteria. If C5 would be exposed to these photons, as its pigment profile suggests, this could lead to more photodamage. Consistently, this population displayed higher expression of proteins like thioredoxin, for scavenging reactive oxygen species, and orange carotenoid protein for photoprotection (Fig. 4).

Inorganic carbon fixation and acquisition are central to realizing high primary productivity and the associated enzymes were highly expressed. The rate-limiting, Calvin-Benson-Bassham Cycle (CBB) enzyme RuBisCO accounted for ~1% of the expressed proteomes of cyanobacterial MAGs, a large fraction for a single enzyme (Fig. 4). In contrast, the expression of the carbon concentrating mechanism (CCM, needed for bicarbonate uptake) varied greatly among cyanobacteria. In C1 and C8, CCM proteins accounted for less than 0.2% of the proteomes. In C5,

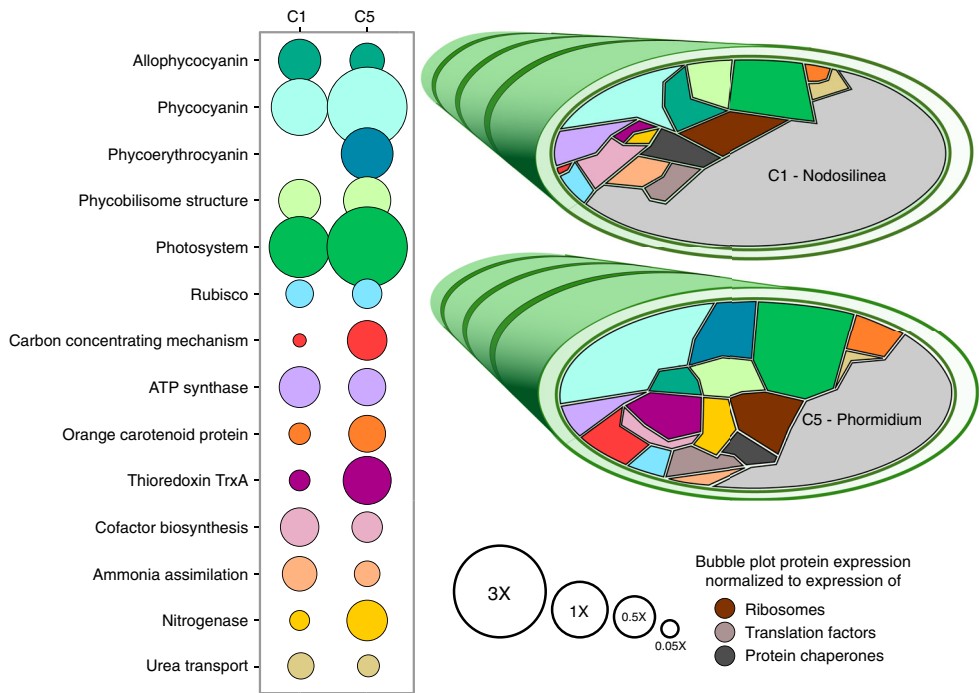

**Fig. 4** Protein expression in the most abundant filamentous cyanobacteria. Bubble plot and Voronoi diagrams comparing expression levels of functions by MAGs C1 and C5, both associated with filamentous cyanobacteria. The area of each shape in the Voronoi diagram is proportional to the percent that protein or subsystem accounts for out of the MAG's expressed proteins. Colour scheme of the Voronoi diagram is the same as the bubble plot. Size of the bubble in the bubble plot is normalized against the relative abundances of ribosomal proteins, translation factors, and protein chaperones in the MAG's proteome. See also Supplementary Data 6

CCM proteins accounted for almost 3% of the expressed proteomes. C5 was the only population to express CCM proteins to a greater level than RuBisCO proteins, suggesting that this population might, to some extent, deplete bicarbonate in its micro-environment. Indeed, C5's $\delta^{13}C$ value was $-20.6 \pm 2.7$‰, compared to $-25.2 \pm 0.8$‰ for C1. A decrease in isotopic fractionation during photosynthesis is usually associated with $CO_2$ (or bicarbonate) limitation[49]. We might conclude that C5's access to higher energy radiation leads to a higher rate of photosynthesis, increased oxygen production, a higher need for protection against free radicals, a higher growth rate and a need for active import of bicarbonate. At a relative abundance of up to 2.3%, C5 was not the most abundant cyanobacterium, so if it had a higher growth rate, it must also have had a higher decay rate. This would make this organism an ecological R strategist, prioritizing cell growth over cell conservation. Because of the high dissolved bicarbonate concentration in these lakes (Supplementary Table 1), it is unlikely that bicarbonate was persistently limiting growth. It is more likely that limitation occurred occasionally, in thick mats or after dilution of dissolved bicarbonate after rain or snow melt.

**Proteomics consistent with low nutrient concentrations**. Nitrogen is a commonly limiting nutrient for primary production in soda lakes globally[50]. The Cariboo Plateau lakes also display low or undetectable concentrations of ammonium and nitrate in lake waters (Supplementary Table 1). Consistently, no expression was detected for any proteins involved in nitrogen loss processes, such as nitrification or denitrification, or for assimilatory nitrate reductases or nitrate transporters.

Many bacteria, including the cyanobacteria C1, C5, and C8, expressed the key genes for the energetically expensive process of nitrogen fixation (Fig. 3, Supplementary Data 6). All

cyanobacteria further expressed glutamine synthetase, for the assimilation of ammonia under nitrogen limiting conditions[51], and the urea transporter. Dinitrogen, urea and, possibly, ammonia, were apparently the main nitrogen sources supporting photosynthesis. Parallel performance of nitrogen fixation by different bacteria provided functional redundancy, contributing to functional robustness and resiliency.

Phosphate can also be a limiting nutrient in soda lakes[50], and this appeared to be the case for Deer Lake in the present study, where phosphate was undetectable in lake waters (Supplementary Table 1). Cyanobacterium C8 (*Gloeocapsa*) was the most abundant population in Deer Lake (12.9% of Deer Lake metaproteome), and expressed a high-affinity phosphate transport system at higher levels (1.5% of C8 expressed proteome) than the other cyanobacteria. Phosphate potentially limited primary production in Deer Lake, as anoxygenic photoheterotrophs were 4–40× more abundant here than in the other lakes (Fig. 3, Supplementary Data 2 and 6).

**Diversity of phototrophs in lake mats**. The microbial mats of the Cariboo region display steep oxygen and sulfide gradients[6], providing opportunities for photoheterotrophic bacteria that use any remaining light, which penetrates beyond the oxic layer created by cyanobacteria[38,52]. Puf or Puh photosystem reaction center proteins were expressed by purple non-sulfur bacteria affiliated with Rhodobacteraceae, MAG A4, and Geminicoccales, MAG A7, as well as autotrophic purple sulfur bacteria, affiliated with *Thiohalocapsa*, MAG G8. Both photoheterotrophs were relatively abundant in phosphate-limited Deer Lake, at 3.2% and 2.8%, respectively. In addition to *puhA*, MAG A4 expressed all three subunits of carbon monoxide dehydrogenase (*coxSML*). Carbon monoxide could be produced by photooxidation of organic material[53], and could serve as an alternative energy source for

these bacteria. Organic substrates supporting photoheterotrophic growth likely consist of cyanobacterial fermentation products, glycolate from photorespiration[38] or could originate from bio-mass decay. By re-assimilation of organic matter or re-fixation of bicarbonate using light energy, these organisms enhance the overall productivity of the mats.

Most unexpected among photoheterotrophs was population Ge1, a representative of an uncultured family within the recently defined phylum Gemmatimonadota. This particular population expressed the *pufC* subunit of the photosynthetic reaction center and contains the remaining photosystem genes in its genome (*pufLMA*, *puhA*, *acsF*). The ability for members of this phylum to use light energy was only recently discovered[54], and the capacity for phototrophy appears to be widespread among members of that phylum[55].

The Gemmatimonadetes bacterium isolated by Zheng and colleagues is heterotrophic, without evidence for a carbon fixation pathway. Interestingly, all genes required for a complete carbon-fixing CBB cycle are present in the genome of MAG Ge1. Genes homologous to the functional RuBisCO Form 1 C large subunit (*rbcL*), and RuBisCO small subunit (*rbcS*) were identified, as well as a copy of the CBB cycle-specific enzyme Phosphoribulokinase (*prk*). These genes were arranged sequentially in the genome: *rbcS*, *rbcL*, and *prk*, an arrangement that points at facultative autotrophy[56]. Upon further investigation of the published MAGs from the Kulunda Steppe soda lakes in Central Asia, we found five additional Gemmatimonadetes MAGs (Fig. 5), that encoded these three CBB cycle genes with the same synteny, and with 88–98% amino acid identity, to the genes of Ge1. All identified *rbcL* genes are functional Form 1 C *rbcL* sequences (Fig. 5b). To our knowledge no other sequenced representatives from the Gemmatimonadetes phylum, apart from these six MAGs, contain the full suite of CBB cycle genes. Given the large number of

amino acids (>90%) shared with homologuous genes encoded in Alphaproteobacteria (e.g., Rhizobiales bacterium YIM 77505 *rbcL*), it seems likely that the last common ancestor of these Gemmatimonadetes populations acquired the CBB genes via horizontal gene transfer from an Alphaproteobacterium, prior to the dispersal and speciation of the clade into the Kulunda Steppe and Cariboo Plateau populations. Although assembly and binning of genomes from metagenomic data sometimes lead to artefactual inference of a horizontal gene transfer event, detection of six sets of phylogenetically congruent genes in six different MAGs from two independent datasets, is unlikely to be artifact. We did not detect expression for these genes and were not able to estimate the $\delta^{13}C$ value for this bacterium (too few high quality MS1 spectra) so it remains unknown to what extent this bacterium used bicarbonate as a carbon source.

**Sulfur cycle in lake mats identified in proteomes.** The presence of the autotrophic purple sulfur bacterium G8, affiliated with *Thiohalocapsa*, indicated active sulfur cycling within the mats, as expected based on the previous detection of sulfide within the mats[6]. Indeed, MAG D1, affiliated with *Desulfonatronum*[20,57] expressed *aprAB*, *sat*, and *dsrAB*, indicating that at least part of the sulfide was produced inside the mats. It also expressed an alcohol dehydrogenase, a formate dehydrogenase, and a hydro-genase, indicating that it oxidized compounds such as ethanol, formate, and hydrogen. These could be derived from dark fer-mentation by cyanobacteria or from decaying biomass. Sulfide produced by D1 was likely re-used by MAGs G8 and G4, the latter affiliated with *Thioalkalivibrionaceae*[18,58]. G4 expressed *soxX*, *soxC*, *dsrA*, and *fccB*, suggesting sulfide oxidation through both the sox pathway and the reverse dsr pathway. Expression of *sox* and *fcc* was also detected for other unbinned populations,

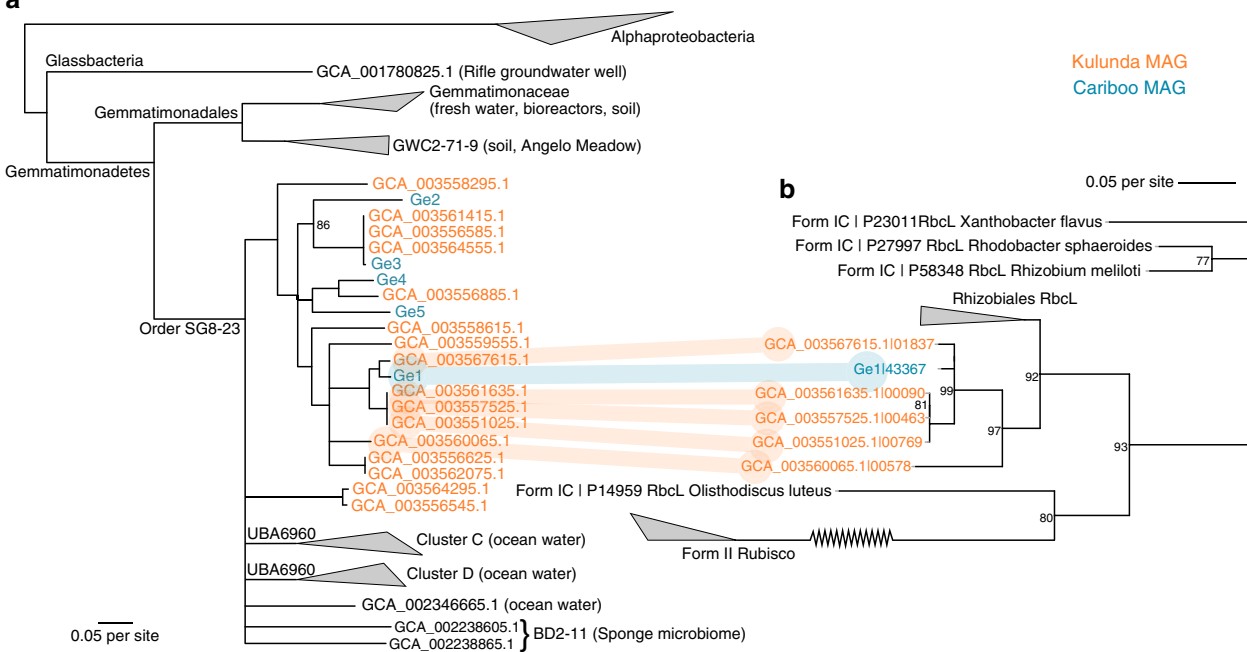

**Fig. 5** Lateral gene transfer of RuBisCO into Gemmatimonadetes phototrophs. **a** Maximum likelihood phylogenetic tree of 19 concatenated ribosomal and RNA polymerase genes obtained from Gemmatimonadota MAGs available in public databases, as well as five MAGs obtained from Cariboo lakes (blue, Ge1–Ge5) and sixteen from Kulunda lakes (orange). Soda lake MAGs form two clusters within GTDB order SG8-23, which further contains several clusters previously recovered from marine habitats. **b** Maximum likelihood phylogenetic tree of RuBisCO Form 1 encoded on MAGs in one of the Gemmatimonadota clusters. The congruence between the two trees indicates vertical inheritance after a single horizontal gene transfer event from Alphaproteobacteria. Nodes with <75% bootstrap support were collapsed into multifurcations. For remaining nodes, bootstrap values were 100% unless indicated otherwise. Supplementary Tables 3 and 4 contain the list of reference sequences used for Fig. 5a, b

affiliated with Alphaproteobacteria, Chromatiales, and other Gammaproteobacteria.

In conclusion, we used metaproteomes and metagenomes to address fundamental questions on the microbial ecology of soda lake mats. We obtained 91 metagenome-assembled-genomes and showed that part of these taxa define a core microbiome, a group of abundant bacteria present in all samples over space (four lakes) and time (4 years). We showed that a very similar community assembled independently in Central Asian soda lakes. The similarity between some of the microbial genomes found in these soda lake regions, incredible in the light of their vast physical separation, suggests that vectors for dispersal are generally ineffective, but can sometimes distribute abundant community members at the global scale. We also showed both functional redundancy and existence of complemental niches among cyanobacteria, with evidence for K and R strategists living side by side. Cyanobacterium C1 was always most abundant but appeared to grow more slowly than C5, based on expression and isotopic signatures. C5 appeared to grow sufficiently fast to occasionally deplete bicarbonate in its surroundings, inconsistent with the prevailing paradigm of unlimited access to bicarbonate in alkaline soda lakes. The nature and origin of carbon sources for photoheterotrophs, including potentially mixotrophic Gemmatimonadetes is an exciting avenue for future research. The presented core microbiome provides a blueprint for design of a productive and robust microbial ecosystem that could guide effective biotechnology for carbon dioxide conversion.

## Methods

**Study site and sample collection**. Samples from benthic microbial mats were collected from four lakes in the Cariboo Plateau region of British Columbia, Canada in May of 2014, 2015, 2016, and 2017. Microbial mats from Last Chance Lake, Probe Lake, Deer Lake, and Goodenough Lake were sampled (coordinates from Supplementary Table 1). Mats were homogenized, immediately frozen, transported on dry ice, and stored at −80 °C within 2 days of sampling. In 2015 and 2017, water samples for aqueous geochemistry were also taken and stored at −80 °C until analysis.

**Aqueous geochemistry**. Frozen lake water samples were thawed and filtered through a 0.45 µm nitrocellulose filter (Millipore Corporation, Burlington, MA) prior to analysis. Carbonate/bicarbonate ($HCO_3^-$) alkalinity analysis was conducted using an Orion 960 Titrator (Thermo Fisher Scientific, Waltham, MA), and concentrations were calculated via double differentiation using EZ 960 software. Major cations ($Ca^{2+}$, $Mg^{2+}$, $K^+$, and $Na^+$) were analyzed using a Varian 725-ES Inductively Coupled Plasma Optical Emission Spectrophotometer (ICP-OES). Major anions ($Cl^-$, $NO_3^-$, $PO_4^{3-}$, and $SO_4^{2-}$) were analyzed using a Dionex ICS 2000 ion chromatograph (Dionex Corporation, Sunnyvale, CA), with an Ion Pac AS18 anion column (Dionex Corporation, Sunnyvale, CA).

Water for reduced nitrogen quantification was filtered through a 0.2 µm filter (Pall Life Sciences, Port Washington, NY). Concentrations were measured using the ortho-phthaldialdehyde fluorescence assay[59], with excitation at 410 nm, and emission at 470 nm.

**Amplicon sequencing and data processing**. DNA extraction and amplicon sequencing were performed, with primer sets TAReuk454FWD (565 f CCAGCA SCYGCGGTAATTCC) and TAReukREV3 (964b ACTTTCGTTCTTGATYRA), targeting Eukaryota, and S-D440 Bact-0341-a-S-17 (b341, TCGTCGGCAGCGTC AGATGTGTATAAGAGACAGCCTACGGGAGGCAGCAG), and S-D-Bact-0785-a-A-21 (805 R, GTCTCGTGGGCTCGGAGATGTGTATAAGAGA-CAGGACTA CHVGGGTATCTAATCC) targeting Bacteria[10]. Sequencing was performed using the MiSeq Personal Sequencer (Illumina, San Diego, CA) using the 2 × 300 bp MiSeq Reagent Kit v3. The reads were processed with MetaAmp[60]. After merging of paired-end reads (>100 bp overlap and <8 mismatches in the overlapping region), primer trimming and quality filtering (<2 mismatches in primer regions and at most 1 expected error), trimming to 350 bp, reads were clustered into operational taxonomic units (OTUs) of >97% sequence identity. Non-metric multidimensional scaling (NMDS) was performed in R, using the package *vegan*[61]. For NMDS, OTUs <1% abundant in all samples were excluded, as were those affiliated with Metazoa, because of large variations in rRNA copy and cell numbers.

**Shotgun metagenome sequencing and data processing**. Metagenomes of the 2015 mat samples were sequenced[62]. Briefly, DNA was sheared into fragments of

~300 bp using a S2 focused-ultrasonicator (Covaris, Woburn, MA). Libraries were created using the NEBNext Ultra DNA Library Prep Kit (New England Biolabs, Ipswich, MA) according to the manufacturer's protocol, which included a size selection step with SPRIselect magnetic beads (Beckman Coulter, Indianapolis, IN) and PCR enrichment (eight cycles) with NEBNext Multiplex Oligos for Illumina (New England Biolabs, Ipswich, MA). DNA concentrations were estimated using qPCR and the Kapa Library Quant Kit (Kapa Biosystems, Wilmington, MA) for Illumina. 1.8 pM of DNA solution was sequenced on an Illumina NextSeq 500 sequencer (Illumina, San Diego, CA) using a 300 cycle (2 × 150 bp) high-output sequencing kit at the Center for Health Genomics and Informatics in the Cumming School of Medicine, University of Calgary. Raw, paired-end Illumina reads were filtered for quality[63]. After that, the reads were coverage-normalized with BBnorm (sourceforge.net/projects/bbmap) with target = 100 min = 4. Overlapping reads were merged with BBMerge with default settings. All remaining reads were assembled separately for each library with MetaSpades version 3.10[64], with default parameters. Contigs of <500 bp were not further considered. tRNA, ribosomal RNA, CRISPR elements, and protein-coding genes were predicted and annotated using MetaErg (sourceforge.net/projects/metaerg/). Per-contig sequencing coverage was estimated and tabulated by read mapping with BBMap, with default settings and "jgi_summarize_bam_contig_depths", provided with MetaBat[65]. Each assembly was binned into metagenome-assembled-genomes (MAGs) with MetaBat with options "-a depth.txt –saveTNF saved_2500.tnf –saveDistance saved_2500.dist -v –superspecific -B 20–keep". MAG contamination and completeness was estimated with CheckM[66]. MAGs were classified with GTDBtk (version 0.2.2, database release 86)[36], together with MAGs previously obtained from Kulunda soda lakes[30]. fastANI was used to compare MAGs across libraries/assemblies[67]. Relative sequence abundances of MAGs were estimated as (MAG contig sequencing coverage) × (MAG genome size) / (total nucleotides sequenced). 16 S rRNA gene sequences were obtained with Phyloflash2[68] and were associated with MAGs based on phylogeny and sequencing coverage covariance across samples, and to OTUs based on sequence identity (Supplementary Table 2). Core genes of MAG variants were identified using blast and these genes were used to determine the abundances of variants across samples using BBMap, with parameters minratio = 0.9 maxindel = 3 bwr = 0.16 bw = 12 fast ambiguous = toss. To identify diversified core genes, variants were aligned with mafft[69] and only genes with >50 single nucleotide polymorphisms (SNPs), >1% of positions with a SNP, and with a fraction of non-synonymous SNPs of >0.825 were kept.

**Protein extraction and metaproteomics**. Protein was extracted and analyzed from 2014 mat samples[62]. Briefly, lysing matrix bead tubes A (MP Biomedicals) containing mat samples and SDT-lysis buffer (0.1 M DTT) in a 10:1 ratio were bead-beated in an OMNI Bead Ruptor 24 for 45 s at 6 m s⁻¹. Next, tubes were incubated at 95 °C for 10 min, spun down for 5 min at 21,000 × g and tryptic peptides were isolated from pellets by filter-aided sample preparation (FASP)[70]. Peptides were separated on a 50 cm × 75 µm analytical EASY-Spray column using an EASY-nLC 1000 Liquid Chromatograph (Thermo Fisher Scientific, Waltham, MA) and eluting peptides were analyzed in a QExactive Plus hybrid quadrupole-Orbitrap mass spectrometer (Thermo Fisher Scientific). Each sample was run in technical quadruplicates, with one quadruplicate run for 260 min with 1 µg of peptide loaded, and the other three for 460 min each, with 2–4 µg of peptide loaded.

Expressed proteins were identified and quantified with Proteome Discoverer version 2.0.0.802 (Thermo Fisher Scientific), using the Sequest HT node. The Percolator Node[71] and FidoCT were used to estimate false discovery rates (FDR) at the peptide and protein level respectively. Peptides and proteins with DFR >5% were discarded. Likewise, proteins without protein-unique-peptides, or <2 unique peptides were discarded. Relative protein abundances were estimated based on normalized spectral abundances[72]. Abundances of MAGs in the metaproteome were estimated by dividing the sum of the relative abundances for all of its expressed proteins by the sum of the relative protein abundances for all expressed proteins. The identification database was created using predicted protein sequences of binned and unbinned contigs, after filtering out highly similar proteins (>95% amino acid identity) with cd-hit[73], while preferentially keeping proteins from binned contigs. Sequences of common contaminating proteins were added to the final database (http://www.thegpm.org/crap/), which is available under identifier PXD011230 in ProteomeXchange. In total, 3,014,494 MS/MS spectra were acquired, yielding 298,187 peptide spectral matches, and 7217 identified proteins. Per population stable isotope fingerprints were estimated based on spectra obtained for all samples[41].

**Phylogenetic analysis**. For the MAG phylogenetic tree (Fig. 5), a set of 16 ribosomal genes[74] plus the RNA polymerase genes rpoABC (TIGR02013, TIGR02027, TIGR02386) were identified and aligned as previously described[74]. After removing poorly aligned regions with gblocks (75, used with option "−b5 = h"), the alignments were concatenated (5053 positions total) and bootstrapped maximum likelihood phylogeny was estimated with RaxML, using model PROTGAMMALG, as described in ref. [74]. All Gemmatimonadota genomes present in GTDB were included as reference sequences. Supplementary Table 3 shows all reference sequences used as well as their geographical origin. The RuBisCO tree was made in the same manner as the MAG phylogenetic tree (636 positions), and RuBisCO

reference sequences can be found in Supplementary Table 4. Expanded Gemmatimonadetes tree can be found in Supplementary Fig. 1 and the expanded RuBisCO tree can be found in Supplementary Fig. 2.

**Reporting summary**. Further information on research design is available in the Nature Research Reporting Summary linked to this article.

## Data availability

Amplicon sequences can be found under the Bioproject PRJNA377096. The 16 S rRNA sequence Biosamples are: SAMN06456834, SAMN06456843, SAMN06456852, SAMN06456861, SAMN09986741-SAMN09986751, and the 18 S rRNA sequence Biosamples are: SAMN09991649-SAMN09991660. The metagenome raw reads and metagenome-assembled-genomes can also be found under the Bioproject PRJNA377096. The Biosamples for the metagenome raw reads are SAMN10093821-SAMN10093824, and the Biosamples for the MAGs are SAMN10237340-SAMN10237430. The metaproteomics data have been deposited to the ProteomeXchange Consortium via the PRIDE partner repository[76] with the dataset identifier PXD011230.

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

## Acknowledgements

We thank the University of Calgary's Center for Health Genomics and Informatics for sequencing and informatics services. We thank Michael Nightingale and Agasteswar Vadlamani for help with analysis of aqueous geochemistry. We also thank Timber Gillis, Hayley Todesco, Harsimrit Lakhyan, Zachary Urquhart, Oliver Horanszky, Virginia Hermanson, Christopher Chow, Peter Zhao, Tong Wang, and Sydney Urschel for help with sample collection and DNA extractions. We would like to thank Dan Liu and Angela Kouris for help with metaproteomics sample preparation and analysis. We thank Carmen Li for help with MiSeq sequencing, and Maryam Ataeian for help with meta-genome analysis. This study was supported by the Natural Sciences and Engineering Research Council (NSERC), Canada Foundation for Innovation (CFI), Canada First Research Excellence Fund (CFREF), Genome Canada, Western Economic Diversification, the International Microbiome Center (Calgary), Alberta Innovates, the Government of Alberta, and the University of Calgary.

## Author contributions

J.Z. collected samples, analyzed data, made figures, and wrote manuscript. C.S. conceived study, collected samples, extracted DNA, and prepared libraries for sequencing. M.K. extracted protein, performed metaproteomics, and analyzed data. P.G. and R.P. performed metagenomics sequencing. X.D. analyzed data, wrote manuscript, and developed pipelines used in metagenomics data analysis. M.S. conceived study, analyzed data, made figures, and wrote manuscript. All authors provided feedback to the manuscript.

## Additional information

**Competing interests:** The authors declare no competing interests.

