## [Peer Review File · Nature Communications]

Reviewers' comments:

Reviewer #1 (Remarks to the Author):

Overall this study is interesting and the manuscript is well written. In it they leverage metagenomic reconstruction to obtain a core set of genotypes from distinct soda lakes. They also used proteomics to examine the activity of these genomes in nature.

Ln 101 – You call it contamination, but CheckM is just identifying duplicate genes? This doesn't really mean they are contaminants, unless you were to look at them in more detail and truly say they are. I suggest referring to them as "gene duplications".

Figure – I find it difficult to assess how similar the genomes generated here are to those, which have been characterized in the past. For example "Ge1-5" is one of the new MAG obtained in the study correct? This is not clear from reading the legend. How similar is Ge1-5 to the order SG8-23? SG8-23 is an order? What is this from SG8-23? Is this something that has been seen in alkaline lakes before? It would things much clearer to include a full tree of this as a supplementary figure.

Figure 5 – A) I would not trust a phylogenetic tree generated from GTDBtk, which uses Fasttree2. There are novel deeply branched clades in this tree that are often results inaccurate branching orders. Clearly the closely related lineages are consistent, but their associations with references at the family level are not reliable. Also, this tree doesn't not contain any statistical support values. B) How was this phylogeny generated? Again there's not bootstrapping support.

Reviewer #2 (Remarks to the Author):

The paper by Zorz et al performs the metagenomic and metaproteomic diversity analysis of microbial communities distributed across distant soda lakes. Briefly, the authors performed the assembly of over 90 metagenomes belonging to the most abundant bacteria observed during a 4 year period to be shared across lakes. In addition, the authors performed metaproteomics analysis and from the over 7000 expressed proteins could conclude that phototropic populations occupied complementary niches. Moreover they identified Gemmatimonadota as a potential photoheterotrophic lineage containing a complete carbon-fixing CBB cycle.

This paper provides valuable insights into the microbial physiology and taxonomic distribution of soda lake communities and shows similarities between microbial communities present in geographical distant soda lakes.

I only have one concern dealing with the phylogenetic analysis and comparisons based in figure 5. In order to do tree comparisons, bootstrap supports should be presented as well as indications regarding the chosen model, type of phylogenetic reconstruction etc. In the Methods section only the software used is given. I would suggest the use of IqTree or similar for maximum likelihood reconstructions instead of MEGA.

In addition, inclusion in the main text of references to the low contamination of the Gemmatimonadota genomes discussed should be provided to support the HGT scenario given for Rubisco.

Reviewer #3 (Remarks to the Author):

Zorz et al report community composition in productive mat communities from alkaline soda lakes. The study reports amplicon, metagenome, and proteome sequencing from four lakes and spans four years of data. The authors report a core microbiome of bacteria that span the samples and note that their data aligns with other studies of alkaline soda lakes. From the protein data, the authors describe resource allocation in phototrophic mats. Finally, the authors report a potentially photoautotrophic members of the Gemmatimonadetes.

The manuscript presents significant new data on alkaline soda lakes, including genomes from metagenomes and the proteomics highlighting phototrophic members of the mats. I find the paper to be reasonably well-written and the data and analyses to be sound other than a few comments and clarifications requested below.

For the most part, I particularly enjoyed the figures. However, for Figure 1, I think more information could be provided with the lakes - for instance where were samples collected and what is the distribution of the mats at each lake? Also, can you please provide pictures of the mats for visualizing the distribution of layers and thickness. This is of interest too regarding how the samples were collected and homogenized / selected for nucleic acid and protein extraction.

There is a lot of data included in the present study which I appreciate. To increase readability of the results and discussion, I think it would be useful to include headers for subsections of the discussion.

The eukaryotic data is not really highlighted (which is okay) — was that the intention? Figure 1 and highlights the difference in abundance of eukaryotes and particularly the phototrophic Eukaryotes in Last Chance Lake (line 91) but this is not re-visited and is not discussed when considering the core microbiome or other aspects of the interpretation of the data. For instance, is it notable that the core microbiome is similar despite a shift in phototrophs from bacteria to eukarya?

Other specific comments:

Abstract:

Line 28: How do you relate reads to total DNA? Can you clarify (perhaps in the main text - lines 102-105). I'm a bit confused on how this calculation works without knowing complete genome sizes of all the microbes recovered.

Introduction:

Line 61: Are these really "whole" genome sequences? Does this imply complete or just the presence of single copy marker genes?

Line 70: What are poor in calcium — basalts? This sentence structure leaves the meaning confusing.

Results and Discussion:

Line 87: Diverse compared to what? Is this species richness? And at what % OTU level? Can you clarify this statement to address these issues.

Line 101: MAGs needs to be outside of the parentheses here somewhere.

Line 108: See comment above - how were species assigned to the amplicon data?

Line 167-168: So the mats do not deplete inorganic carbon locally? Has this been demonstrated somewhere?

Lines, 232-233, 235-240: This interpretation implies inorganic carbon varies in the mat which is inconsistent with what is written above (e.g. Line 167-168) and throughout where the text implies the system is always replete with inorganic carbon.

Lines 253: I don't see methods for phosphate. Phosphate tends to be really low when lots of biological activity is going on, even in eutrophic lakes. Did you digest the sample water prior to running analyses correct? If not, its unlikely you detected any dissolved organic P (unless the P was loosely bound). Therefore, I am skeptical about the low levels of P.

281: More accurate to say that the genes are present in the genome (rather than possession)

Line 302: Fix reference

Line 305: Is there evidence for sulfide production in the sediments below the mats or are you suggesting it is all produced in the mat?

Line 321: Can you discuss this idea more explicitly in the main text? The idea for R strategists is clear but perhaps it could be made more clear the K vs. R strategists from your data set.

Methods:

Is there any reason to expect differences in water chemistry for the years not reported? Particularly rainy season or heavy snowpack, etc.?

Please provide methods for the PO₄³⁻ data/

Please add / provide detection limits for water chemistry data.

Was DNA extracted from the total mat? Were they sectioned at all? How thick was the mat?

Figure 3: How is "Abundance in the Metagenome" calculated and how is that comparable to "Abundance in the Metaproteome"? I can see that MAG abundance was based on contig sequence coverage but what about individual protein sequences or metabolic pathways?

Point by point responses to the reviewers' comments

We would first like to thank the reviewers for taking the time to read our manuscript and for providing valuable feedback. Below we have addressed each point made by the reviewers and highlighted these responses in blue. Where applicable, we have supplied the changes made to the text to address these comments below our responses. We have uploaded a new Figure 1 with sampling locations and images of the microbial mats. We have also uploaded a new Figure 5 that has been re-made from scratch, with precise MAFFT alignments and true, bootstrapped maximum likelihood analyses, as suggested by the reviewers. A new section on the protocol used to generate the phylogenetic trees has been added to the methods section. We have also updated Supplementary table 1 with detection limits, and added Supplementary table 8 (16S amplicon OTUs matched with MAGs), Supplementary table 9 (Reference sequences for Gemmatimonadetes phylogenetic tree), and Supplementary table 10 (Rubisco reference sequences for Rubisco phylogenetic tree). Lastly, we have reduced the abstract to fit within the 150 word limit.

Reviewer #1 (Remarks to the Author):

Overall this study is interesting and the manuscript is well written. In it they leverage metagenomic reconstruction to obtain a core set of genotypes from distinct soda lakes. They also used proteomics to examine the activity of these genomes in nature.

1.1 Ln 101 – You call it contamination, but CheckM is just identifying duplicate genes? This doesn't really mean they are contaminants, unless you were to look at them in more detail and truly say they are. I suggest referring to them at "gene duplications".

We have changed "contaminants" to "duplicated conserved single-copy genes" in the following sentence (Ln 100) and in supplementary table 3:

Line 100: "We selected 91 representative, de-replicated MAGs for further analysis (**Supplementary Table 3**). Most of these MAGs were near-complete (>90% for 85 MAGs) and contained relatively few duplicated conserved single-copy genes (<5%, for 83 MAGs)."

1.2 Figure 5 – I find it difficult to assess how similar the genomes generate here are to those, which have been characterized in the past. For example "Ge1-5" is one of the new MAG obtained in the study correct? This is not clear from reading the legend. How similar is Ge1-5 to the order SG8-23? SG8-23 is an order? What is this from SG8-23? Is this something that has been seen in alkaline lakes before? It would things much clearer to include a full tree of this as a supplementary figure.

We apologize for the lack of clarity. These trees have been entirely redone, based on the comments below, as described in the updated methods section. The revised legend reads:

“Figure 5 – a. Maximum likelihood phylogenetic tree of 19 concatenated ribosomal and RNA polymerase genes obtained from Gemmatimonadota MAGs available in public databases, as well as five MAGs obtained from Cariboo lakes (blue, Ge1 - Ge5) and sixteen from Kulunda lakes (orange). Soda lake MAGs form two clusters within GTDB order “SG8-23”, which further contains several clusters previously recovered from marine habitats. b. Phylogenetic tree of RuBisCO Form 1 encoded on MAGs in one of the Gemmatimonadota clusters. The congruence between the two trees indicates vertical inheritance after a single horizontal gene transfer event from Alphaproteobacteria. Nodes with <75% bootstrap support were collapsed into multifurcations. For remaining nodes, bootstrap values were 100% unless indicated otherwise. Supplementary Tables 9 and 10 contain the list of reference sequences used for Figures 5a and 5b.”

The newly added methods section reads:

Line 429:

“Phylogenetic analysis

For the MAG phylogenetic tree (Figure 5), a set of 16 ribosomal genes (74) plus the RNA polymerase genes rpoABC (TIGR02013, TIGR02027, TIGR02386) were identified and aligned as previously described (74). After removing poorly aligned regions with gblocks (75, used with option “-b5=h”), the alignments were concatenated (5053 positions total) and bootstrapped maximum likelihood phylogeny was estimated with RaxML, using model PROTGAMMALG, as described in (74). All Gemmatimonadota genomes present in GTDB were included as reference sequences.

Supplementary Table 9 shows all reference sequences used as well as their geographical origin. The Rubisco tree was made in the same manner as the MAG phylogenetic tree, and Rubisco reference sequences can be found in **Supplementary Table 10**. Expanded Gemmatimonadetes tree can be found in **Supplementary Figure 1** and the expanded Rubisco tree can be found in **Supplementary Figure 2.**”

Two new references were added:

“74. Hug LA, Baker BJ, Anantharaman K, Brown CT, Probst AJ, Castelle CJ, Butterfield CN, Hermsdorf AW, Amano Y, Ise K, Suzuki Y, Dudek N, Relman DA, Finstad KM, Amundson R, Thomas BC, Banfield JF (2016) Nature Microbiol 1, 16048

75. Talavera G, Castresana J (2007) Improvement of phylogenies after removing divergent and ambiguously aligned blocks from protein sequence alignments. Systematic Biol 56, 564-577.”

1.3 Figure 5 – A) I would not trust a phylogenetic tree generated from GTDBtk, which uses Fasttree2. There are novel deeply branched clades in this tree that are often results inaccurate branching orders. Clearly the closely related lineages are consistent, but their associations with references at the family level are not reliable. Also, this tree doesn't not contain any statistical support values.

B) How was this phylogeny generated? Again there's not bootstrapping support.

We thank the reviewer for pointing out our lack of thorough phylogenetic analysis. We have redone this analysis from scratch, see response to comment 1.2 above. Note that the branching order

remained unchanged, but some nodes were shown to have low support and were collapsed into multifurcations.

Reviewer #2 (Remarks to the Author):

The paper by Zorz et al performs the metagenomic and metaproteomic diversity analysis of microbial communities distributed across distant soda lakes.

Briefly, the authors performed the assembly of over 90 metagenomes belonging to the most abundant bacteria observed during a 4 year period to be shared across lakes.

In addition, the authors performed metaproteomics analysis and from the over 7000 expressed proteins could conclude that phototropic populations occupied complementary niches. Moreover they identified Gemmatimonadota as a potential photoheterotrophic lineage containing a complete carbon-fixing CBB cycle.

This paper provides valuable insights into the microbial physiology and taxonomic distribution of soda lake communities and shows similarities between microbial communities present in geographical distant soda lakes.

2.1 I only have one concern dealing with the phylogenetic analysis and comparisons based in figure 5. In order to do tree comparisons, bootstrap supports should be presented as well as indications regarding the chosen model, type of phylogenetic reconstruction etc. In the Methods section only the software used is given. I would suggest the use of IqTree or similar for maximum likelihood reconstructions instead of MEGA.

We thank the reviewer for pointing out our lack of thorough phylogenetic analysis. We have redone this analysis from scratch, see response to comment 1.2 above.

2.2 In addition, inclusion in the main text of references to the low contamination of the Gemmatimonadota genomes discussed should be provided to support the HGT scenario given for Rubisco.

We thank the reviewer for pointing out that this should not be taken for granted. We believe that the strength of the evidence for lateral gene transfer in this case mainly lies in the detection of these genes in six different MAGs, rather than in the low degree of “contamination” of a single MAG (~5% for Ge1). We have added a sentence to the main text as follows:

Line 296: “Although assembly and binning of genomes from metagenomic data sometimes leads to artefactual inference of a horizontal gene transfer event, detection of six sets of phylogenetically congruent genes in six different MAGs from two independent datasets, is unlikely to be artifact.”

Reviewer #3 (Remarks to the Author):

Zorz et al report community composition in productive mat communities from alkaline soda lakes. The study reports amplicon, metagenome, and proteome sequencing from four lakes and spans four years of data. The authors report a core microbiome of bacteria that span the samples and note that their data aligns with other studies of alkaline soda lakes. From the protein data, the

authors describe resource allocation in phototrophic mats. Finally, the authors report a potentially photoautotrophic members of the Gemmatimonadetes.

The manuscript presents significant new data on alkaline soda lakes, including genomes from metagenomes and the proteomics highlighting phototrophic members of the mats. I find the paper to be reasonably well-written and the data and analyses to be sound other than a few comments and clarifications requested below.

3.1 For the most part, I particularly enjoyed the figures. However, for Figure 1, I think more information could be provided with the lakes - for instance where were samples collected and what is the distribution of the mats at each lake? Also, can you please provide pictures of the mats for visualizing the distribution of layers and thickness. This is of interest too regarding how the samples were collected and homogenized / selected for nucleic acid and protein extraction.

We thank the reviewer for pointing out these omissions. We have indicated sampling locations in Figure 1 and added extra panels with pictures from the mats. The mats were about 1 cm thick and had quite different appearances. They were homogenized before extraction of DNA and protein. We have updated the methods section to make the sampling procedure more clear:

(Line 328): “Samples from benthic microbial mats were collected from four lakes in the Cariboo Plateau region of British Columbia, Canada in May of 2014, 2015, 2016, and 2017. Microbial mats, approximately 1 cm thick, from Last Chance Lake, Probe Lake, Deer Lake, and Goodenough Lake were sampled (coordinates in Supplementary Table 1). Mats were homogenized, immediately frozen, transported on dry ice, and stored at -80°C within 2 days of sampling. In 2015 and 2017, water samples for aqueous geochemistry were also taken and stored at -80°C until analysis.”

The revised legend to Figure 1 reads: “**Figure 1** – Satellite images of **A** Deer Lake, **B** Goodenough and Last Chance Lakes, **C** Probe Lake. Black stars indicate approximate sampling locations. **D**. Bubble plots showing the relative contribution of Cyanobacteria and Eukaryotes to the lake metaproteomes. **E**. Non-metric multidimensional scaling (NMDS) plots using Bray-Curtis dissimilarity to visualize the microbial communities of the soda lake mats over years of sampling using 16S rRNA amplicon sequencing data, and **F**. 18S rRNA amplicon sequencing data. Shapes indicate year of sampling: Circles: 2014, square: 2015, diamond: 2016, triangle: 2017. Samples for 18S rRNA analysis were not taken in 2016, and Deer Lake samples were not taken in 2014 for 18S, and 2016 for 16S. NMDS Stress values were below 0.11. **G**. Image of microbial mat from Goodenough Lake, and **H**. Image of microbial mats along the shore of Last Chance Lake.”

3.2 There is a lot of data included in the present study which I appreciate. To increase readability of the results and discussion, I think it would be useful to include headers for subsections of the discussion.

Unfortunately, the journal does not permit headers in the discussion section (<https://www.nature.com/documents/ncomms-manuscript-checklist.pdf>)

3.3 The eukaryotic data is not really highlighted (which is okay) — was that the intention? Figure 1 and highlights the difference in abundance of eukaryotes and particularly the phototrophic

Eukaryotes in Last Chance Lake (line 91) but this is not re-visited and is not discussed when considering the core microbiome or other aspects of the interpretation of the data. For instance, is it notable that the core microbiome is similar despite a shift in phototrophs from bacteria to eukarya?

Unfortunately, the larger size and complexity of Eukaryotic genomes challenges current metagenomic approaches. We simply assembled too little evidence to create meaningful interpretations for this habitat's Eukaryotes. Therefore, even though these organisms are most likely extremely interesting in their own right, they will require dedicated studies. We have added a sentence to the revised manuscript about the detection of the core microbiome in Last Chance Lake, as follows:

Line 90: "Probe, Deer and Goodenough Lakes harbored predominantly cyanobacterial mats, whereas the mats of more saline Last Chance Lake contained mainly phototrophic Eukaryotes. This was shown with proteomics (see below), because it was impossible to compare abundances of Eukaryotes and Bacteria using amplicon sequencing. Bacterial species associated with 340 OTUs were found in all four lakes. These species accounted for 20.5% of the region's species richness and 84% of the total sequenced reads, suggesting that there is a common and abundant "core" microbiome shared among the alkaline lakes of the Cariboo Plateau. **Despite the high proportion of eukaryotic biomass, and phototrophs, the core alkaline lake, prokaryotic microbiome was still present in Last Chance Lake (although at lower relative abundances).**"

Other specific comments:

Abstract:

3.4 Line 28: How do you relate reads to total DNA? Can you clarify (perhaps in the main text - lines 102-105). I'm a bit confused on how this calculation works without knowing complete genome sizes of all the microbes recovered.

Indeed, the statement was elaborated on lines 102-105. The calculation might be more simple than the reviewer assumes. It does not even require an estimate of genome size for individual MAGs. We simply divided the total number of reads mapped to the 91 MAGs by the total number of reads sequenced. Assuming Illumina sequencing is not biased toward our MAGs (a reasonable assumption given their wide range of GC content), this proportion would also apply to extracted DNA. To avoid jargon, we opted for "extracted DNA", rather than "sequencing reads". Please note that estimated genome sizes are provided in supplementary table 3 and are necessary to estimate abundances of individual populations. See also response to comment 3.21 for calculations of abundances of individual MAGs.

Introduction:

3.5 Line 61: Are these really "whole" genome sequences? Does this imply complete or just the presence of single copy marker genes?

To avoid a discussion on the “completeness” of MAGs at this point, we have modified this sentence to:

Line 61: “Recently, almost one thousand Metagenome Assembled Genomes (MAGs) were obtained from sediments of Kulunda soda lakes (30).”

3.6 Line 70: What are poor in calcium — basalts? This sentence structure leaves the meaning confusing.

We have changed the sentence to be more clear:

Line 67: “Underlying basalt in some areas of the plateau, originating from volcanic activity during the Miocene and Pliocene eras, provides ideal conditions for forming soda lakes, as it provides little soluble calcium and magnesium.”

Results and Discussion:

3.7 Line 87: Diverse compared to what? Is this species richness? And at what % OTU level? Can you clarify this statement to address these issues.

We thank the reviewer for pointing out this omission. This sentence was modified to:

“Four years of amplicon sequencing data (16S and 18S rRNA) showed the microbial mats contain at least 1,662 bacterial and 587 eukaryotic species-level operational taxonomic units (OTUs, clustered at 97% similarity).”

3.8 Line 101: MAGs needs to be outside of the parentheses here somewhere.

We have changed this sentence to (see also comment 1.1 above):

Line 100: “We selected 91 representative, de-replicated MAGs for further analysis (Supplementary Table 3). Most of these MAGs were near-complete (>90% for 85 MAGs) and contained relatively few duplicated conserved single-copy genes (<5%, for 83 MAGs).”

3.9 Line 108: See comment above - how were species assigned to the amplicon data?

The 16S rDNA amplicon workflow was described in the methods section and more details are provided in reference 59. For example, species were assigned to OTUs by blasting the OTU consensus sequences against the SILVA database. MAGs were classified taxonomically using GTDBtk, as described in the methods section and reference 36. Full length 16S rDNA sequences were obtained with Phyloflash (reference 67) and paired to MAGs based on coverage profiles and taxonomy, as described in the methods section. MAG-full length 16S sequences were paired with OTUs based on perfect sequence overlap, as described in the results section:

“Perfect alignment of full length 16S rDNA gene sequences to consensus OTU amplicon sequences showed that almost all these MAGs were core Cariboo microbiome members, present in each lake.”

For precision, we have modified the sentence on L106 to:

“This was not surprising because amplicon sequencing had already indicated the presence of >2,000 different bacterial and eukaryotic OTUs.”

We have also added **Supplementary Table 8** which shows the assignment of OTUs to MAGs.

3.10 Line 167-168: So the mats do not deplete inorganic carbon locally? Has this been demonstrated somewhere?

Indeed, this was shown for these lakes in reference 6, as cited. Brady et al (2013) measured the isotopic biosignatures of the microbial mats present in these same soda lakes. They concluded that the isotopic biosignatures of the mats as a whole, showing significant fractionation against dissolved inorganic carbonates, were “consistent with non-CO₂-limited photosynthesis”.

For clarity, we have modified this sentence to:

Line 166-168: But, to what extent do these known “rules of engagement” also apply to alkaline soda lake microbial mats, where primary productivity has access to unlimited inorganic carbon (2), as was previously shown for Cariboo soda lakes (6)?

3.11 Lines, 232-233, 235-240: This interpretation implies inorganic carbon varies in the mat which is inconsistent with what is written above (e.g. Line 167-168) and throughout where the text implies the system is always replete with inorganic carbon.

We thank the reviewer for making that important point. Both the high expression of bicarbonate importers and the lower isotopic fractionation point in the direction of bicarbonate limitation for this organism. Still, the observed fractionation was a respectable 20 per mill. In the light of the high bicarbonate concentrations (tens of mmol/L) complete/persistent limitation is highly unlikely. We have modified the text to clarify this point, and added some possible explanations:

Line 230: “In **C5**, CCM proteins accounted for almost 3% of the expressed proteomes. **C5** was the only population to express CCM proteins to a greater level than RuBisCO proteins, suggesting that this population might, to some extent, deplete bicarbonate in its micro-environment. Indeed, **C5**’s $\delta^{13}\text{C}$ value was $-20.6 \pm 2.7\text{‰}$, compared to $-25.2 \pm 0.8\text{‰}$ for **C1**. A decrease in isotopic fractionation during photosynthesis is usually associated with CO₂ (or bicarbonate) limitation (48). We might conclude that **C5**’s access to higher energy radiation leads to a higher rate of photosynthesis, increased oxygen production, a higher need for protection against free radicals, a higher growth rate and a need for active import of bicarbonate. At a relative abundance of up to 2.3%, **C5** was not the most abundant cyanobacterium, so if it had a higher growth rate, it must also have had a higher decay rate. This would make this organism an ecological R strategist, prioritizing cell growth over cell conservation, at least compared to **C1**. Because of the high dissolved bicarbonate concentration in these lakes (Supplementary table 1), it is unlikely that bicarbonate was persistently limiting growth. It is more likely that limitation occurred occasionally, in thick mats or after dilution of dissolved bicarbonate after rain or snow melt.”

3.12 Lines 253: I don't see methods for phosphate. Phosphate tends to be really low when lots of biological activity is going on, even in eutrophic lakes. Did you digest the sample water prior to running analyses correct? If not, its unlikely you detected any dissolved organic P (unless the P was loosely bound). Therefore, I am skeptical about the low levels of P.

We thank the reviewer for pointing out that omission (see also comment 3.18). We only measured inorganic phosphate, not organic phosphate. Methods for phosphate have been updated in the methods section:

Line 342: "Major anions (Cl^- , NO_3^- , PO_4^{3-} and SO_4^{2-}) were analyzed using a Dionex ICS 2000 ion chromatograph (Dionex Corporation, Sunnyvale, CA), with an Ion Pac AS18 anion column (Dionex Corporation, Sunnyvale, CA)"

3.13 281: More accurate to say that the genes are present in the genome (rather than possession)

We have changed the sentence to say that "the genes are present in the genome":

Line 281: "Interestingly, all genes required for a complete carbon-fixing CBB cycle are present in the genome of MAG Ge1"

3.14 Line 302: Fix reference

We have fixed the reference (see also response to comment 3.15 below). We also noticed an error in another reference and fixed this as well:

Line 192: "Proteomic data were also used to estimate the ^{13}C content of some abundant species, providing additional information on which carbon source they used and to what extent their growth was limited by carbon availability (74)."

3.15 Line 305: Is there evidence for sulfide production in the sediments below the mats or are you suggesting it is all produced in the mat?

Indeed, Brady et al (reference 6) previously measured sulfide profiles using micro-electrodes, showing evidence for the presence of sulfide in the mats. For clarity, we have rephrased as follows:

"The presence of the autotrophic purple sulfur bacterium **G8**, affiliated with *Thiohalocapsa*, indicated active sulfur cycling within the mats, as expected based on the previous detection of sulfide within the mats (6). Indeed, MAG **D1**, affiliated with *Desulfonatronum* (20,56) expressed *aprAB*, *sat*, and *dsrAB*, indicating that at least part of the sulfide was produced inside the mats."

3.16 Line 321: Can you discuss this idea more explicitly in the main text? The idea for R strategists is clear but perhaps it could be made more clear the K vs. R strategists from your data set.

See response to comment 3.11, for additional text explaining K and R strategies.

Methods:

3.17 Is there any reason to expect differences in water chemistry for the years not reported? Particularly rainy season or heavy snowpack, etc.?

As these lakes experience large changes in water level each year (from high, after snowmelt in Spring, to low at the end of summer), we expect large differences, mainly during each year, but also between years, caused by variation in precipitation. How specific microbial populations and the microbial community as a whole adapt to these changes would require a much larger sampling effort. Our study shows, that despite these differences, each lake maintains a distinct community, and overall, at least the core microbiome is resilient to these changes.

3.18 Please provide methods for the PO43- data/

Thank you for catching this omission. Phosphate was measured in the same way as the other anions. The methods have been updated accordingly (see response to comment 3.12).

3.19 Please add / provide detection limits for water chemistry data.

We have added the detection limits for the water chemistry data as a new column in Supplementary Table 1.

3.20 Was DNA extracted from the total mat? Were they sectioned at all? How thick was the mat?

Yes, DNA was extracted from the total mat. Total mat thickness was about 1 cm. The mats are very fragile, heterogeneous and meaningful sectioning would be challenging. This information has been added to the methods section for clarification (see response to comment 3.1).

3.21 Figure 3: How is “Abundance in the Metagenome” calculated and how is that comparable to “Abundance in the Metaproteome”? I can see that MAG abundance was based on contig sequence coverage but what about individual protein sequences or metabolic pathways?

We thank the reviewer for pointing out the lack of clarity. We have modified the methods section, from

Line 393: “Relative sequence abundances of MAGs were estimated based on contig sequencing coverage.”

to

“Relative sequence abundances of MAGs were estimated as (MAG contig sequencing coverage) x (MAG genome size) / (total nucleotides sequenced).”

For abundance in the metaproteome, from

Line 420: "Relative protein abundances were estimated based on normalized spectral abundances (72)."

to

"Relative protein abundances were estimated based on normalized spectral abundances (72). Abundances of MAGs in the metaproteome were estimated by dividing the sum of the relative abundances for all of its expressed proteins by the sum of the relative protein abundances for all expressed proteins."

and added a sentence to the legend of Figure 3:

"MAG abundances in metagenome and metaproteome were estimated as explained in Materials and Methods."

REVIEWERS' COMMENTS:

Reviewer #1 (Remarks to the Author):

All of my comments have been addressed.

Reviewer #2 (Remarks to the Author):

In this revised version the authors addressed all of reviewers comments including recomputing the phylogenetic analyses with better methods.

Minor: please change the legend of figure 5b to Maximum Likelihood phylogeny and indicate how many positions were used for this reconstruction.

Reviewer #3 (Remarks to the Author):

The manuscript presents significant new data on alkaline soda lakes, including genomes from metagenomes and the proteomics highlighting phototrophic members of the mats. I find the revised paper addresses my original comments, concerns, and suggested edits.

REVIEWERS' COMMENTS (NCOMMS-19-13113):

Reviewer #1 (Remarks to the Author):

All of my comments have been addressed.

Reviewer #2 (Remarks to the Author):

In this revised version the authors addressed all of reviewers comments including recomputing the phylogenetic analyses with better methods.

Minor: please change the legend of figure 5b to Maximum Likelihood phylogeny and indicate how many positions were used for this reconstruction.

We have changed the legend of figure 5b to:

"b. Maximum Likelihood Phylogenetic tree of..."

636 positions were used for the alignment of the RuBisCO sequences:

"The RuBisCO tree was made in the same manner as the MAG phylogenetic tree (636 positions), and RuBisCO reference sequences..."

Reviewer #3 (Remarks to the Author):

The manuscript presents significant new data on alkaline soda lakes, including genomes from metagenomes and the proteomics highlighting phototrophic members of the mats. I find the revised paper addresses my original comments, concerns, and suggested edits.